# Mathematical Methods Applied to the Problem of Dairy Cow Replacements: A Scoping Review

**DOI:** 10.3390/ani15070970

**Published:** 2025-03-27

**Authors:** Osvaldo Palma, Lluis M. Plà-Aragonés, Alejandro Mac Cawley, Víctor M. Albornoz

**Affiliations:** 1Department of Mathematics, Universidad de Lleida, 73 Jaume II, 25001 Lleida, Spain; lluismiquel.pla@udl.cat; 2Department of Economics and Administration, Universidad Nacional Andrés Bello, Santiago 7500000, Chile; 3Agrotecnio CERCA Center, 191, Rovira Roure, 25198 Lleida, Spain; 4Department of Industrial and Systems Engineering, Pontificia Universidad Católica de Chile, Santiago 7820436, Chile; amac@ing.puc.cl; 5Department of Industrial Engineering, Campus Santiago Vitacura, Universidad Técnica Federico Santa María, Santiago 7650568, Chile; victor.albornoz@usm.cl

**Keywords:** dairy, milk, replacement, optimization, simulation, machine learning

## Abstract

This study reviews the mathematical models used to address the problem of dairy cow replacements, which is crucial for the economic performance of dairy farms. The review highlights that dynamic programming is the most commonly used optimization technique, followed by stochastic and deterministic simulations. Machine learning methods are still rarely applied in this field. The primary focus of these methods is on optimizing milk production and profitability. This research underscores the importance of these mathematical approaches in enhancing the efficiency, profitability, and sustainability of dairy farming operations.

## 1. Introduction

The problem of cow replacements is economically important for the dairy and beef industry, and mathematically complex. Replacement policies strongly impact the financial performance of a herd, because replacement heifers represent a significant fraction of a farm’s total costs [1]. Selecting the wrong time for replacements implies economic losses, either because the cow has not yet achieved her productive potential, or on the contrary, it is old and has started to decline in milk production, while maintaining or increasing production costs (e.g., feed and veterinary costs). In this sense, Kalantari et al. [2] indicated that dairy cow replacements are crucial for obtaining optimal economic value, influenced by milk production, pregnancy status, cow health, lactation stage, and replacement animal value.

One factor that makes dairy cow replacements highly complex is that cows differ from other livestock species (e.g., special factors such as performance variability in reproduction) [3]. In addition, given the multiperiod character and the biological uncertainty involved, proposed mathematical approaches to the problem have been complex as well. For instance, some works report different methods to solve the replacement problem based on Markov decision processes or dynamic programming. However, the use of these methods to tackle real instances makes the ‘curse of dimensionality’ a problem [1,4,5]. Adding new features to herd models exponentially increases the state space, making it difficult to generate practical models, as pointed out by Cabrera [6], who reported a low level of real-life applications of models developed until 2010. In contrast, Heikkilä et al. [7] indicated that few solutions had added disease effects to replacement models at the time of their publication. Monti et al. [8] highlighted that culling decisions are important for herd performance and cannot be studied separately from economic considerations, animal health, and management practices. This naturally implies generating increasingly complex mathematical models [9].

We have found three reviews related to our topic, the most recent conducted 15 years ago by Stygar and Makulska [10], who analyzed 20 publications on mathematical models in beef herd management. They had a broader scope, with replacement being only a system component, classifying models as either optimization or simulation models. These 20 articles were not included in our work since replacement was not in the center. The review by Monti et al. [8] focused on defining and understanding culling causes through 15 publications. Only one reference fell within our review’s scope and was included [11]. Finally, the review by Lehenbauer and Oltjen [12] considered mathematical models for the replacement problem. However, we found these shortcomings: (i) it is an old work published 25 years ago, when computational capacity was an important issue for solving real complex problems, (ii) they mentioned optimization methods without considering simulation models, (iii) machine learning tools were not available at that time, and (iv) this review did not mention a systematic review method, nor the process followed to obtain the revised papers. Our work identified twenty articles published before 1998; however, we have selected only three of them [13,14,15] for this latest revision. Our review presents an updated work, which focuses on the specific topic of dairy cow replacement, and which includes the full field of mathematical models applied to this topic, complementing the search with modern machine learning methods not considered in previous reviews. Given the above, we are interested in exploring how mathematical methods have been used in the problem of replacing dairy cows through a scoping review. We aim to identify the different mathematical tools used, the areas of application (individual cow or complete herd), the response variables, the economic indicators, the types of production (beef or milk), and the extensive type (barn or grazing). This information will help us understand the gaps in this topic, allowing us to propose new characteristics for future models and suggest new techniques that can significantly contribute to this field. To achieve this, we formulate the following research question: how have mathematical tools been applied to the problem of dairy cow replacements? We aim to identify what has been conducted so far and future research opportunities. Relevant studies will be selected through the scoping review method, followed by a descriptive data analysis.

## 2. Materials and Methods

We have carried out a scoping review following the PRISMA-ScR statement [16]. Scoping reviews are systematic bibliographical search processes whose results help investigate the knowledge and evidence about a specific topic as well as answer broad research questions. This methodology allows us to determine theories, fundamental concepts, and knowledge gaps around a topic of interest. It is the latter aspect that is the most relevant for our research purpose. The application of mathematical methods to dairy cow replacement problem has been seldomly studied, even less if we consider the recent rise of new machine learning tools. Therefore, scoping reviews can detect knowledge gaps of our topic. In this way, our research domain is adequate for performing a scoping review because studies regarding the application of optimization, simulation, and machine learning to the replacement of dairy cows are seldomly explored. To achieve our goal, we followed a standard scoping study procedure comprising five steps: (1) identify the research questions; (2) identify relevant studies; (3) identify selection criteria; (4) chart the data; and (5) collate, summarize, and report the results.

### 2.1. Research Questions

Following the methodology of a scoping review, to start the research and with the aim of focusing the search, a broad research question is posed that aims to be the basis of our study. This general question is as follows: how have mathematical tools been applied to the replacement problem on dairy farms? However, given the amplitude of the subject and the comprehensive sources of the reports, we break down the main research question into five more specific research questions:

RQ1. Is the scope of the models presented in the selected studies aimed at the entire herd or that of an individual cow?

RQ2. Is the main type of production of the herd milk or beef?

RQ3. Is the type of herd extension grazing or stable?

RQ4. What are the mathematical methodologies used in the selected studies to solve the dairy cow replacement problem?

RQ5. What are the response variables of the models? What is the main economic indicator used in the models?

To answer these questions, we developed a rigorously structured and sufficiently documented method to provide robust evidence and arguments.

### 2.2. Identify Relevant Studies

In our scoping review, we searched the Web of Science and Scopus database for all relevant studies using keywords that we consider the best representatives of the study objective. The words used were dairy, milk, cow, herd, cattle, replacement, optimization, simulation, and machine learning. These words were entered into the search engines, considering the combinations of the Boolean operators available in the search engines that would allow us to obtain a small number of studies whose results were effectively related to the objective of the study. The specific keywords entered in both search engines are as follows: ((dairy OR milk OR cow OR herd OR cattle) AND (replacement) AND (optimization OR simulation OR machine learning)). After we obtained the results in both search engines, the available filters mentioned below were applied: (i) article and review for document type, and (ii) English for language.

### 2.3. Selection Criteria

The definition of different inclusion and exclusion criteria was determined post hoc, because the researchers’ familiarity with the studies increased. In the first exclusion process (screening), we considered only articles published in peer-reviewed journals. In the next exclusion step, the paper titles and keywords were individually verified to determine their alignment with the research topic. The remaining articles that passed the screening were then checked for abstracts’ alignment with the research topic: how have mathematical tools been applied to the problem of replacing dairy cows? Subsequently, the full texts of the selected results were reviewed. Those that were not consistent with the study objectives or papers with the text body not written in English were eliminated. The study selection process was carried out by the first author and supervised by the other co-authors. Authors are aware of sources of bias related to the choice of articles written in English, the keywords that authors have entered into the WOS and Scopus search engines, and the subjective criteria in the rejection of borderline papers.

## 3. Results and Discussion

### 3.1. Identify Relevant Studies and Selection Criteria

The results of our selection process of relevant articles are summarized in Figure 1. Finally, 40 selected articles were obtained. The scientific articles found are included in Table 1. In the search criteria, a filter for the years of the publications was not considered, covering both old and recent works from 1979 to 2024. Entering the selected words into the search engines yielded 429 publications in WOS and 501 in Scopus. After removing 238 duplicates from the two search engines, 692 publications remained. Subsequently, based on titles, keywords, and abstracts, we eliminated 641 publications. For example, the work entitled ‘Gross margin losses due to Salmonella Dublin infection in Danish dairy cattle herds estimated by simulation modelling’ [17] was excluded because it is not related to mathematical tools applied to the problem of dairy cow replacements. Another example of deletion due to the abstract not being aligned with our topic was the work of [18], which considered the replacement problem but did not focus on mathematical methods. With the remaining 51 publications, we reviewed the full texts and eliminated 11 more for not being related to the main topic (reason 1). For example, [19] was excluded because the study focused on comparing organic and conventional milk production systems and did not directly address the problem of replacing dairy cows using mathematical tools. Thus, 40 publications were obtained. The search for articles in WOS and Scopus was carried out during the month of December 2024.

These 40 publications were evaluated, thanks to the implementation of broad search terms and a search period that spanned from the first publications registered in the search engines to the year 2024. The search strategy was carried out in WOS and Scopus databases to provide comprehensive coverage of dairy-related research, although it is likely that some publications located in other databases were not included in the analysis. For the selection of publications relevant to this study, a scoping review was conducted with the aim of effectively answering our research questions. Although the result of 40 publications can be considered low due to the importance of the topic, the number is the result of a systematic search and selection method described above. We have performed an independent search in Scopus by entering the words (dairy OR milk OR cow OR herd OR cattle AND replacement) and we obtained 10,095 results, a high number that was reduced to 26 if “cow replacement” or “heifer replacement” was used in the search. This fact confirmed most cases do not concentrate on the application of mathematical methods nor on the replacement problem.

### 3.2. Chart the Data

We used the Bibliometrix (https://www.bibliometrix.org/home/) [49] and Vosviewer [50] software to perform a bibliometric analysis. With these tools, we have a simpler approach to perform different analyses and discover research networks in the context of dairy production.

In a preliminary approach, we can highlight that revised articles are on average 24.1 years old since their year of publication. There were 15 journals containing the 40 selected papers, averaging around 2.7 publications per journal. The average number of authors per article is 3.2 and the number of articles that were made by a single author is 5, which is a small number with regards to all the selected papers.

In the word cloud of articles’ keywords (Figure 2), keywords were related to milk production: replacement (seven times), dairy cow (five times), dairy cattle (four times), and calving pattern (three times); other keywords were related to the effort to optimize the value of livestock: dynamic programming (11 times), economic (seven times), optimization (seven times), simulation (six times), and Markov chain (two times).

Leveraging the tools provided by VOSviewer software (version 1.6.2), Figure 3 indicates the relationships between the authors’ keywords in the 40 selected articles. This image shows us naturally that the strongest relationship is generated by the keywords cattle, cow, herds, model, replacement, and replacement policies. These concepts are generally linked to the other keywords. Some features of Figure 3 that we can highlight are as follows: (i) dynamic programming is the primary method used to perform replacement policy optimization; (ii) insemination is an important term that interacts with other terms such as replacement policies, optimization, simulation, and dynamic programming; (iii) simulation is also a main term that connects with several others such as calving pattern, performance, and economic; (iv) the term machine learning appears slightly in the figure; and (v) small groups of words related to diets, lactation, or meat production curves and Markov decision processes appear.

In Figure 4, it is noteworthy that from 1979 to 2024, the number of publications related to the application of mathematical tools in the replacement problem in dairy cows is relatively scarce; in many years, no published works were registered, the maximum number of works for a year was three publications, most of the works were registered between 1992 and 2012, and in 2024, two articles were registered again. There is no clear trend in the number of articles over time.

The journals where the selected works were mainly published include *The Journal of Dairy Science* (*n* = 14), *Agricultural Systems* (*n* = 7) and *Livestock Production Science* (*n* = 6), which contain approximately 68% of total publications. There are 11 journals that contain only one publication of the selected papers.

The author with the highest production on the topic is Kristensen AR with seven articles, followed by Cabrera VE (*n* = four), Dijkhuisen AA (*n* = four), Kalantari AS (*n* = four) and van Arendonk JAM (*n* = four).

The main objective of our research work is embodied in the general research question that asks how mathematical tools have been applied to the problem of dairy cow replacements. To facilitate our work, this general research question has been subdivided into five more specific questions. To analyze our results with a broad view, we first turned to the bibliometric tools of Bibliometrix and VOSviewer. Our bibliometric study obtained the groupings and interactions generated between the elements (keywords, authors, affiliations, etc.) present in the 40 selected articles. Figure 4 shows that from 1979 to 2024, the topic of the application of optimization, simulation, and machine learning tools to the problem of dairy cow replacements has been very seldomly studied. There is no clear trend in the number of studies over time, with several years in this period lacking publications. The most prolific years correspond to 2000 and 2012, each presenting no more than three publications. The year 2024 contains two publications on this topic. This indicates that interest in this area of study is low but maintained over time.

The keyword point cloud analysis indicates that the predominant words can be classified into two groups: one related to the object of study (dairy cattle, dairy cows, replacement, calving pattern, and culling) and another which considers the techniques used to carry out those studies (dynamic programming, optimization, simulation, economic, and management). According to the information from these keywords, the most commonly used technique in this field is dynamic programming, followed by the simulation methodology. The aim is to obtain replacement policies that optimize the value of the dairy herd, which is the final objective pursued by these studies.

Reinforcing the previous idea, we can consider Figure 3. The graphical analysis of the keyword grouping allows us to conclude that the concepts of cattle and dairy cows are naturally linked with the other concepts. Other important links include economics and modeling, as well as dynamic programming and simulation. We can interpret that in this area of study, dynamic programming and simulation methods are strongly used to solve the problem of dairy cow replacement. However, it is striking that keywords associated with machine learning methods are not present in a relevant way.

### 3.3. Results Related to Specific Research Questions

In this section, we are dedicated to methodically carrying out the reading and review with more structure and in greater depth than the standard bibliographic reviews carried out for the selected articles.

The analysis of studies on applied mathematical tools to the problem of dairy cow replacements reveals a variety of approaches and results. In terms of the level of application, 95% of the studies focus on the whole herd (Table 2). Regarding the first research question (RQ1), it is known that in an organization, there are three levels of decision-making: strategic, tactical, and operational. The strategic level focuses on long-term goals, overall direction, and resource allocation. Tactical decisions deal with implementing short and medium-term strategies, involving the allocation of specific resources. Finally, operational decisions concentrate on daily actions to ensure effectiveness and efficiency in routine activities. These levels combine to achieve organizational objectives, ensuring coherence between strategy, tactics, and operational execution. The results on the scope of application of optimization, simulation, and machine learning models applied to the problem of dairy cow replacements show that 95% of the studies focus on the entire herd. However, several of these studies consider the herd from a macro point of view, developing their models by describing the dynamics of a single animal. This characteristic can be observed in both optimization models (e.g., Cabrera [6]) and simulation models (e.g., Clausen et al. [31]). Herd-level studies use models such as Markov chains and stochastic simulation to capture complex interactions and evaluate management policies that affect the entire herd, thus optimizing overall productivity and profitability. This strategic approach allows decisions to be made that impact the entire system.

Among the optimization techniques used, dynamic programming stands out, being used in 58% of the studies (Table 2). Other optimization methodologies include Markov decision processes (15%), multilevel Markov hierarchical processes (3%), and Markov chains (8%). On the other hand, the simulation method is present in 48% of the cases. Stochastic simulation corresponds to 40% of the total 40 articles, while deterministic simulation is present in 8% of the cases. Machine learning methods applied to the replacement problem of dairy cows are only present in two articles, representing 5%. Six studies use both optimization and simulation methods.

Related to the fourth research question (RQ4), the predominance of dynamic programming in 58% of the studies reflects its effectiveness and flexibility in addressing complex optimization problems in the management of dairy cow replacements. This technique allows decisions to be modeled and multiple states and actions to be considered, which is crucial for optimizing livestock productivity [43]. The characteristic that most of the selected works use dynamic programming is due to the fact that these models are of a discrete and non-continuous type; this is because milk production depends on reproductive processes that are essentially discrete [51]. The presence of Markov decision processes and Markov chains, although in smaller proportions (15% and 8%, respectively), indicates their usefulness in modeling state transitions and decisions under uncertainty, providing an important basis for strategic decision-making (e.g., [5,35]). The simulation technique, present in 48% of the studies, highlights the importance of this methodology in evaluating different scenarios and management strategies. Stochastic simulation, used in 40% of cases, is particularly valuable for incorporating the variability and uncertainty inherent in dairy production, allowing researchers and managers to test various policies and observe their potential impacts. Deterministic simulation, although less common (8%), is still relevant for scenarios where conditions are more predictable and controlled. The preference for stochastic simulation over deterministic simulation may be due to the application of stochastic models to generate more realistic models [52].

The limited use of machine learning methods (5%) suggests that, although these techniques have great potential to improve accuracy and personalization in decision-making, their application in this field is still in the early stages. Moreover, the combination of optimization and simulation methods in some studies (15%) reflects an integrated approach that leverages the strengths of both methodologies to provide more robust and comprehensive solutions.

The most common response variable included in the models of the 40 selected articles is milk production, which appears in the 58% of the studies (Table 3). Other response variables that are included in the models presented in the selected articles are diverse, for example the case of cow lifespan (5%) and feed intake (10%), and several other cases that appear in a single publication such as the case of the cost of nitrogen excretion.

In terms of economic indicators, the studies mainly use the benefit indicator in 78% of cases (Table 3). Other economic indicators present in the selected studies are as follows: retention value (RPO, which represents the expected benefit of keeping a cow compared to its immediate replacement) with 8% of cases, net present value (NPV) in 10% of cases, cow value, 3%, future net worth (FP), 3%, economic efficiency, 3%, and annualized net income, 3%.

Regarding RQ5, at least 58% of the studies consider milk production as a response variable, identifying it as the most important response variable in this study. This is due to its direct impact on the profitability and sustainability of dairy farms. According to Kalantari et al. [2], milk production is one of the main variables considered in culling decisions. Finally, our results highlight that the benefit indicator is the most used economic indicator, which is used in 78% of the studies. This may be because it is a direct and simple measure of dairy profitability. Concepts such as net present value are less used, possibly due to the conceptual complexity that requires including a discount rate and using the method of the value of money over time, which is a more complex concept.

The main type of production is milk, present in 93% of the studies analyzed (Table 3), the remaining 7% dealing with beef production problems. In addition, the cow management type includes both grazing, 8%, and barn, 13%, while most studies do not report the type of livestock spread.

Now we try to answer the second research question (RQ2). Of the forty studies, thirty-eight are dedicated to the study of dairy production and only two studies consider the study of beef production. According to Mazzetto et al. [53], the research focuses more on dairy cattle than beef cattle because milk production requires intensive daily management and there is a constant demand for dairy products. In addition, milk production has a significant environmental impact, driving the need for detailed studies to improve efficiency and reduce environmental footprints.

Regarding the third research question (RQ3), a small percentage of studies focus on grazing (8%) and a slightly higher percentage on barn management (13%). Most papers do not report the type of extent of the cattle, which could indicate a lower priority in documenting these details in research. This distribution may reflect prevailing practices in the dairy industry, where barn management is more common due to the need to better control cows’ feed, health, and milk production. In addition, barn management allows for greater intensification and specialization in dairy production, which can contribute to greater efficiency and profitability [54]. 

Our review uses only the research published in the WOS and Scopus databases. It is possible but rare that other important works published in other databases had been excluded from our analysis, as well as important articles not written in English.

As most of the studies reviewed consider Holstein cows, the generalization to other dairy cattle breeds may not apply. The particular characteristics of Holstein cows, such as their high milk production, may not be representative of other breeds. In addition, many of the studies are conducted in continental climate zones (e.g., Denmark, Finland, Netherlands), which can also introduce bias with regard to the overall use of the models in tropical or hot climates. Weather affects forage availability, cow health, and welfare, which can vary significantly depending on temperatures and humidity.

Another characteristic of the 40 selected studies is the simplification of the economic variables and context, also assuming an unlimited availability of heifers for replacement. Therefore, proposed models are well-fitted for supporting strategic decisions, but less for tactical–operational ones.

A common input parameter for dairy cows is the maximum number of parities allowed [55], which corresponds with practical rules used in practice. However, a small number of studies reported the number of productive years achieved by the models (e.g., [2] reports that the expected life of the herd is 3.18 years) and at most they refer to the maximum number of parities. Therefore, we cannot know if there are significant differences between the stayability of cows in these models.

In order to compare the results with previous studies, we have tried to broaden our view with other scoping reviews on similar topics. To achieve this, we use the WOS search engine by entering the words (scoping review AND optimization OR simulation OR machine learning) and selecting the title option in the fields, thus obtaining 342 results. Reviewing the titles of these results, we do not find works that carry out reviews of applications of mathematical methods on aspects of food production or economics related to other types of animals.

### 3.4. Gaps in the Literature and Future Outlook

With regard to the fourth research question posed in our work (what are the mathematical methodologies used in the selected studies to solve the dairy cow replacement problem?), we have identified research gaps: (i) There is a need to explore more accurate models using machine learning tools, such as neural networks and decision trees. (ii) The combination of different methods developing synergies is another interesting research line claimed by different authors (e.g., von Rueden et al. [56]). They remark on the complementarity of simulation and ML. When reviewing the level of hybridization between machine learning and simulation tools in the dairy industry, we did not find studies dedicated to this combination of methods. To explore ways in which ML tools are combined with simulation methods, we can review possible applications of these tools in other non-livestock species, even in other situations a little more distant from livestock where decision support systems can be presented with the implementation of the two tools. An example is the development of autonomous vehicles [57]. These approaches are near the development of digital twins and concepts of augmented reality or virtual reality seeking realistic simulation environments [58]. (iii) As mentioned, the implementation of combined replacement models with different animal diseases is both a research opportunity and a mathematical challenge. For example, we could use cases of replacement models combined with the case of twin births [59], events that impact the profitability of a dairy herd, or the consideration of different types of abortions and their repercussions on the value of the herd. In addition to the opportunities that ML generates in the dairy industry, there are also implementation challenges that include integrating data and applying algorithms to embedded data to improve decision-making on dairy farms [60]. Implementation challenges include the quality of datasets and the need for interpretable models for farmers. In addition, the difficulty of integrating data from various sources due to their heterogeneity and lack of communication between systems are recognized. The usability of support systems for producers is key for them to adopt new technologies. These systems need to be easy to use and understand, helping growers make better decisions and better manage their resources [61]. If the systems are intuitive, producers can monitor their operations more efficiently, increasing farm productivity. (iv) During the research, we did not identify any cases in which optimization or simulation models consider the interaction between dairy cows. As a particular case related to this characteristic, Kulkarni et al. [3] studied the economic impact of the limited supply of replacement heifers, considering that in places like Central Europe, there are environmental pressures to reduce the number of dairy cattle, which will ultimately imply a reduction in the number of replacement heifers to maintain the number of cows in the productive stage. In this article, the economic impact of having a reduced and excessive number of replacement heifers was shown. In both cases, the profitability of the herd is affected. Previous work on the limited supply of replacement heifers (e.g., Benari and Gal [23]; Kristensen [26]) considered models in which cows can interact with each other and compete for scarce replacement heifers. However, they do not present the solution to the optimization model due to the high computational requirements of these formulations. (v) In relation to the previous point, a possible option to solve these models with a high level of complexity is to resort to surrogate model techniques. According to Hou and Behdinan [62], surrogate models are useful because they simplify complex engineering processes, such as manufacturing and computer-aided engineering, by replacing full-scale models that are expensive and time-consuming. These models help manage the curse of dimensionality. In addition, they reduce the execution time and memory consumption in highly complex processes [63], using dimensionality reduction algorithms that simplify models without losing significant accuracy.

Mathematical methods applied to dairy cow replacements can influence seemingly distant areas, such as environmental impact and cybersecurity. For instance, the rapid digitization of the dairy sector opens up new risks and introduces security vulnerabilities that had not been considered before, such as cyberattacks, which could compromise critical data and destabilize production [64].

## 4. Conclusions

This article presented a comprehensive review of the application of mathematical tools to the problem of dairy cow replacements, highlighting the importance of these tools in improving the profitability of livestock. Various methodologies and their impact on decision-making were analyzed, emphasizing the relevance of milk production as a key variable, present in 58% of the studies, and economic benefit as the main indicator, used in 78% of the cases. Currently, machine learning tools have been scarcely applied, representing only 5% of the reviewed studies. Additionally, opportunities for future research and the integration of simulation and machine learning techniques were identified, which can contribute to the more effective and profitable management of dairy cattle. Dynamic programming stands out as the most effective technique, but the incorporation of new AI methodologies may lead to further progress in optimizing the efficiency and profitability of dairy farming operations.

## Figures and Tables

**Figure 1 animals-15-00970-f001:**
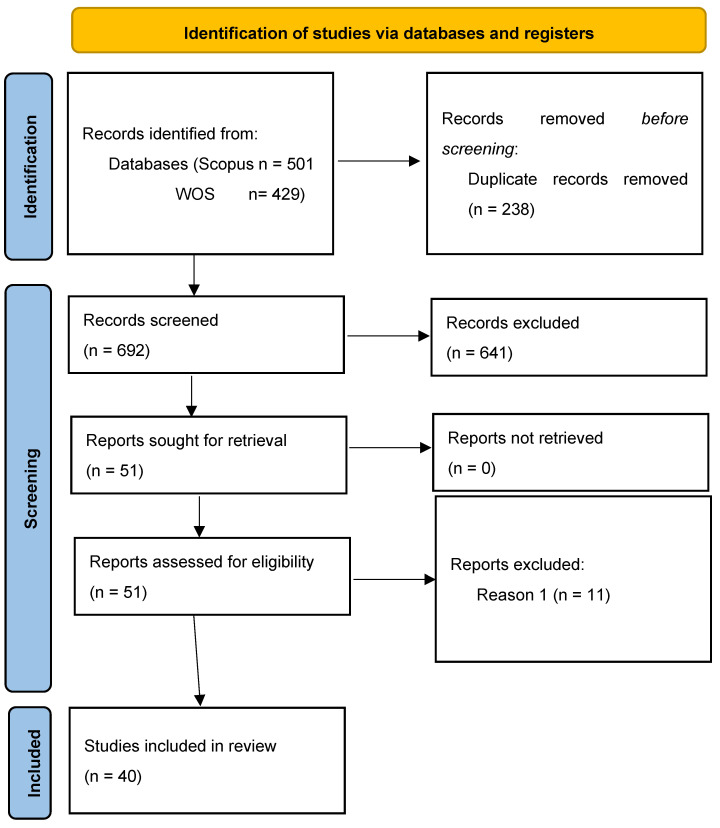
PRISMA protocol flow of the item selection process.

**Figure 2 animals-15-00970-f002:**
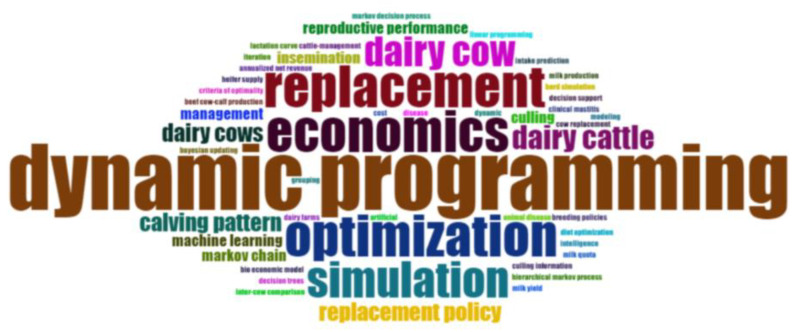
Keyword cloud of the selected articles.

**Figure 3 animals-15-00970-f003:**
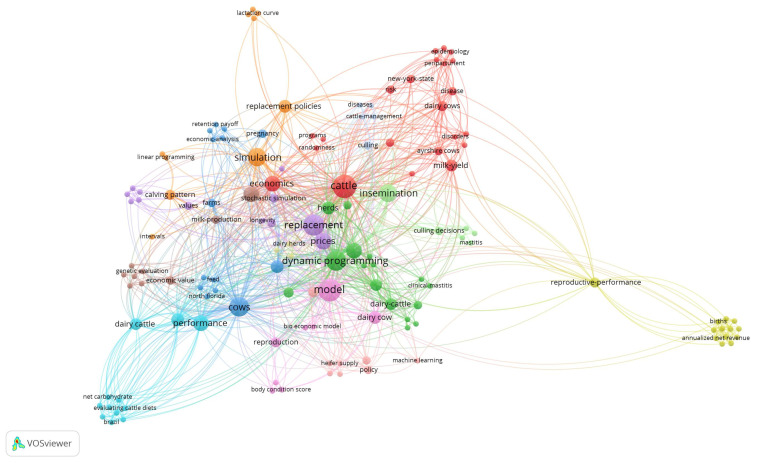
The figure shows the relationship network among the keywords selected by the authors in the 40 publications included in this scoping review.

**Figure 4 animals-15-00970-f004:**
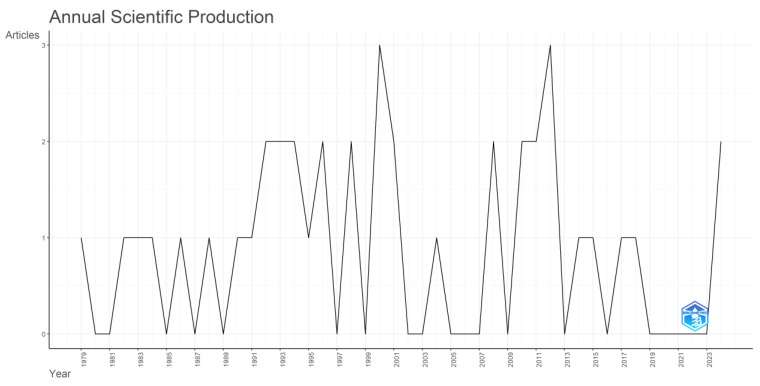
Annual scientific production of the selected articles in our studio.

**Table 1 animals-15-00970-t001:** List of selected articles.

Publication Number in Paper New	Title	Reference
1	Preliminary model to investigate culling and replacement policy in dairy herds	[20]
2	Replacement policy in dairy herds on farms where heifers compete with cows for grassland. 2. Experimentation	[21]
3	Operational replacement decision model for dairy herds	[14]
4	Beef production from a dairy farm a linear programming simulation approach	[22]
5	Optimal replacement policy for multicomponent systems: An application to a dairy herd	[23]
6	An economic comparison of 4 insemination and culling policies in dairy herds, by method of stochastic simulation	[24]
7	Recursive stochastic programming applied to dairy cow replacement	[15]
8	Ranking of dairy cows for replacement	[25]
9	Optimizing model-insemination, replacement, seasonal production, and cash flow	[13]
10	Optimal replacement in the dairy herd a multi component system	[26]
11	Dynamic probabilistic simulation of dairy herd management practices. i. model description and outcome of different seasonal calving patterns	[27]
12	Dynamic probabilistic simulation of dairy herd management practices. ii. comparison of strategies in order to change a herds calving pattern	[28]
13	A survey of markov decision programming techniques applied to the animal replacement problem	[29]
14	Optimizing the herd calving pattern with linear programming and dynamic probabilistic simulation	[30]
15	Technical and economic effects of culling and reproduction strategies in dairy cattle herds estimated by stochastic simulation	[31]
16	Technical and economic effects of feeding one vs. multiple total mixed rations estimated by stochastic simulation under different dairy herd and management characteristics	[32]
17	Evaluation of a stochastic dynamic replacement and insemination model for dairy cattle	[33]
18	Modeling the relationship between reproductive performance and net-revenue in dairy herds	[34]
19	A linear programming formulation of the Markovian decision process approach to modelling the dairy replacement problem	[35]
20	A stochastic model simulating the feeding-health-production complex in a dairy herd	[36]
21	Optimizing replacement decisions for Finnish dairy herds	[37]
22	Optimizing breeding decisions for Finnish dairy herds	[38]
23	Interactions between optimal replacement policies and feeding strategies in dairy herds	[39]
24	Comparison of economically optimized culling recommendations and actual culling decisions of Finnish Ayrshire cows	[40]
25	Economics of delayed replacement when cow performance is seasonal	[11]
26	The cost of generic clinical mastitis in dairy cows as estimated by using dynamic programming	[41]
27	Optimal replacement policy and economic value of dairy cows with diverse health status and production capacity	[7]
28	Determining the optimum replacement policy for Holstein dairy herds in Iran	[2]
29	A large Markovian linear program to optimize replacement policies and dairy herd net income for diets and nitrogen excretion	[6]
30	A multi-level hierarchic Markov process with Bayesian updating for herd optimization and simulation in dairy cattle	[1]
31	Optimal culling strategy in relation to biological and economic efficiency and annualized net revenue in the Japanese Black cow calf production system	[42]
32	The effect of reproductive performance on the dairy cattle herd value assessed by integrating a daily dynamic programming model with a daily Markov chain model	[43]
33	A simple formulation and solution to the replacement problem: A practical tool to assess the economic cow value, the value of a new pregnancy, and the cost of a pregnancy loss	[44]
34	Effects of sex control and twinning on economic optimization of culling cows in Japanese Black cow calf production systems	[45]
35	Short communication: Prediction of retention pay-off using a machine learning algorithm	[46]
36	Stochastic economic evaluation of dairy farm reproductive performance	[47]
37	A robust statistical model to predict the future value of the milk production of dairy cows using herd recording data	[4]
38	Using decision trees to extract patterns for dairy culling management	[48]
39	Enhancing culling decisions in swiss dairy farms: introducing a tool for improved replacement choices	[5]
40	Economic impacts of constrained replacement heifer supply in dairy herds	[3]

**Table 2 animals-15-00970-t002:** Classification of the selected articles based on the type of application (to the herd or to a single cow) and on the mathematical methodology (optimization, simulation, or other).

	Application		Methodology					
			Optimization			Simulation		Other
Paper Number	Herd	Individual	Markov Chains	Markov Decision Process	Linear Programming	Deterministic	Stochastic	
1	X	X					X	
2	X					X		
3	X			X				
4	X				X			
5	X				X		X	
6	X						X	
7		X						Recursive Stochastic Programming
8	X			X			X	
9	X				X			
10	X				X		X	
11	X		X					
12	X				X		X	
13	X			X	X			
14	X				X		X	
15	X	X					X	
16	X						X	
17	X				X			
18	X				X		X	
19	X			X	X			
20	X						X	
21	X				X			
22	X				X			
23	X				X			
24	X				X			
25	X				X		X	
26	X				X		X	
27	X				X			Logistic regression
28	X				X			
29	X				X			
30	X							Multilevel Markov hierarchical process
31	X					X		
32	X		X		X			
33	X	X					X	
34	X					X		
35	X				X			ML decision trees
36	X						X	
37	X							Exponential smoothing, linear regression, and logistic function
38		X						Decision trees
39	X		X					
40	X				X		X	

**Table 3 animals-15-00970-t003:** Classification of the selected articles on the basis of the response variables used, economic indicators, type of production of the herd (beef or milk), and extensive type of production.

	Response Variables			Economic Indicator				Main Production Type		Type of Extensive Production	
Number Publication in Paper	Milk Production	Diseases	Other	Benefit	Income	Cost	Other	Beef	Milk	Grazing	Barn
1	X		Genetic improvement in performance per cow.	X					X	X	
2	X			X					X		X
3							Present net value (NPV)		X		
4			Beef production. Replacement rate. Breed.	X				X			
5							Expected value of keeping a cow in the herd		X		
6	X		Interval between calvings, the rate of discarding, and the rate of detection of estrus and conception.	X					X		
7				X					X	X	
8				X					X		
9	X			X					X		
10									X		
11	X								X		
12	X			X					X		
13			Optimal replacement policies, and herd statistics.				Present net value (NPV)		X		
14	X			X					X		
15	X			X					X		
16				X					X		
17				X					X		X
18				X					X		
19	X			X					X		
20	X		Food intake, food utilization, conception, slaughter, involuntary elimination, and death.	X					X		
21	X			X					X		
22	X		The replacement rate, the structure of the herd.	X					X		
23	X		Average herd lifespan, calving interval, replacement rate, feed intake, and body weight change.	X					X	X	
24	X		Risk of discarding, open days (pregnancy), lactation stage (days in milk), and month of delivery.	X					X		
25				X					X		
26		mastitis				X			X		X
27				X					X		
28	X		Retention value (RPO).	X					X		
29	X		Herd structure, replacement policy, dry matter intake (DMI), and nitrogen excretion (N).	X					X		
30	X		Productive herd life. Interval between calvings. Annual discard rate. Proportion of voluntary discards. Feed intake capacity. Energy requirement. Forage and concentrate intake. Monthly Revenue and Costs. Monthly and lifetime net income.	X					X		X
31			Biological efficiency (BE), the live weight of discarded cows and their calves, metabolizable energy consumption (ME), and economic costs and revenues.				Economic efficiency (EE) and annualized net income (AN).	X			
32			Retention value (RPO)	X					X		
33	X		The value of a new pregnancy, the cost of a pregnancy loss, the pregnancy rate, the discard rate, and the structure of the herd.				Present net value (NPV)		X		
34	X			X				X			
35			Retention value (RPO)	X					X		
36	X		Feed costs, calf sales revenues, discard costs, reproductive costs, herd structure (parity, days in milk, days open), and pregnancy and discard rates.	X					X		
37	X		Somatic cell count, survival, and correlation structures between these herd-level measures.						X		
38	X			X					X		
39	X		Expected lifespan of the cow. Average monthly income per cow. Average monthly costs per cow. State transitions (lactation, month in milk, month of pregnancy). Lactation curves (milk production, protein and fat content).	X			Cow value		X		X
40				X					X		

## Data Availability

Not applicable.

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
