# Peer review of "Mathematical Methods Applied to the Problem of Dairy Cow Replacements: A Scoping Review"

_animals, 2025, doi:10.3390/ani15070970_

Round 1

Reviewer 1 Report

Comments and Suggestions for Authors

The paper provides a comprehensive review of mathematical methods used to optimise dairy cow replacement decisions, a crucial aspect of herd profitability.  
The use of a scoping review to identify optimisation and simulation models provides a structured overview of the existing literature.  
The thesis presents a precise categorisation of the models used, response variables and economic indicators, with clear statistical data.  
The analysis shows that machine learning models are still little explored in this area, suggesting future research opportunities.  
Lines 28-90 contain a lot of general information on the management of substitution in livestock farms, but could be summarised to get to the objectives of the review more quickly. It might be useful to add a paragraph to make the added value of this review over previous work more explicit.  
The study selection process (Section 2.3, lines 125-150) is well described, but a summary table showing the number of articles excluded at each stage could improve the transparency of the method. It would be useful to add a critical analysis of the limitations of the selection, e.g. mentioning the possibility of bias in the selection of articles.  
The review compares the results with some previous studies, but it would be useful to extend the comparison to other systematic and scoping reviews on similar topics. It would be interesting to discuss what new trends are emerging from previous studies and how the development of computational techniques is affecting the field.  
Section 4. Conclusions (lines 378-387) is clear, but the practical implications of the results for farmers and researchers could be explored further.  Including a discussion of how mathematical models can be used in day-to-day farm management would improve the application relevance of the work.  
Lines 339-373 identify machine learning as a research opportunity, but lack a discussion of implementation challenges.  It would be useful to explore the difficulties of integrating ML models with real farm data, such as the quality of available datasets and the need for interpretable models for farmers.  
Here are some specific comments with line numbers
63-85 Avoid repetition of the problem definition and get to the objective of the study more quickly.
112-121 Specify any environmental or market factors that may influence cow replacement decisions.
224-230 Add bibliographical references to better justify the choice of analysis methods.
310-345 Introduce a figure schematising the selection process to improve the transparency of the method.
492-514 Discuss possible practical and computational challenges in implementing optimisation models in real flocks.
Overall, the manuscript has been written with great care and attention; possible improvements include summarising and streamlining the introduction to avoid redundancies; adding a summary table of the item selection process; strengthening the comparison with existing literature to better contextualise the results; and expanding the discussion of the challenges of using machine learning to provide a broader perspective on future research directions.  
The manuscript is of good scientific quality, with a rigorous methodology and well-structured quantitative analysis. The work makes a significant contribution to dairy herd management by identifying future trends and opportunities in cow replacement models.  

Author Response

Dear reviewer, thank you very much for your comments. Your suggestions have provided a valuable opportunity for improvement. I proceed to answer your questions:

Revisor 1

Lines 28-90 contain a lot of general information on the management of substitution in livestock farms, but could be summarised to get to the objectives of the review more quickly. It might be useful to add a paragraph to make the added value of this review over previous work more explicit.  

The introduction to meet this requirement has been summarized (line 36).

The study selection process (Section 2.3, lines 125-150) is well described, but a summary table showing the number of articles excluded at each stage could improve the transparency of the method. It would be useful to add a critical analysis of the limitations of the selection, e.g. mentioning the possibility of bias in the selection of articles.

Lines 146-150 have been included to improve the explanation. Section 3 shows all the results with the number of articles selected and deleted for greater transparency. Possibilities of bias are mentioned.

The review compares the results with some previous studies, but it would be useful to extend the comparison to other systematic and scoping reviews on similar topics. It would be interesting to discuss what new trends are emerging from previous studies and how the development of computational techniques is affecting the field.

On lines 384-390 we respond to this question.

Section 4. Conclusions (lines 378-387) is clear, but the practical implications of the results for farmers and researchers could be explored further.  Including a discussion of how mathematical models can be used in day-to-day farm management would improve the application relevance of the work.  
On lines 423-422 we answer this question.

Lines 339-373 identify machine learning as a research opportunity, but lack a discussion of implementation challenges.  It would be useful to explore the difficulties of integrating ML models with real farm data, such as the quality of available datasets and the need for interpretable models for farmers.  
On lines 423-428 we answer this question.

Here are some specific comments with line numbers 63-85 Avoid repetition of the problem definition and get to the objective of the study more quickly.

The introduction to meet this requirement has been summarized (line 36).

112-121 Specify any environmental or market factors that may influence cow replacement decisions.

On lines 367-373 we answer this question.

224-230 Add bibliographical references to better justify the choice of analysis methods.
On lines 304-307 we answer this question.

310-345 Introduce a figure schematising the selection process to improve the transparency of the method.
On line 189 a figure is given that solves this question.

492-514 Discuss possible practical and computational challenges in implementing optimisation models in real flocks.

On lines 427-432 we answer this question.

Reviewer 2 Report

Comments and Suggestions for Authors

NOTES:
• The paper reviews mathematical models used to solve problems of dairy farming. The review can be used to create appropriate algorithms used in machine learning.
• The review uses only studies published in the WOS and Scopus databases (in the years 1979–2024; see Figure 4).
• The authors devoted Section 3 to methodical reading and reviewing with greater depth than standard bibliographic reviews conducted for selected articles.
• Machine learning can contribute to more effective and profitable dairy farming.
• Review papers organizing previous research results can be very helpful in creating machine learning information systems. Therefore, the work is important and in the current trend of research in applied sciences.
• I suggest publishing the paper. The paper is written carefully and solidly.

SUGGESTED changes:
• Explain in more detail what Figure 2 shows. Maybe it is about Figure 3?
• I suggest adding a comment (e.g. in the summary) about data protection to avoid terrorist attacks on dairy farming.

Author Response

Dear reviewer, thank you very much for your comments. Your suggestions have provided a valuable opportunity for improvement. We proceed to answer your questions:

Revisor 2

Explain in more detail what Figure 2 shows. Maybe it is about Figure 3?

We modified lines 205-209 to answer this question.

I suggest adding a comment (e.g. in the summary) about data protection to avoid terrorist attacks on dairy farming.

On lines 452-456 we answer this question.

Reviewer 3 Report

Comments and Suggestions for Authors
  1. The article focuses on the issue of using mathematical methods to optimize the replacement of dairy cows on dairy farms. The issue is interesting and relatively important from the point of view of farm operation. New findings in this direction could be beneficial for dairy farms all over the world. I have a few comments on the article.
  2. The authors rely on 59 scientific publications listed in the final References in the article. Given the wide range of problems that affect herd turnover, I consider this number to be relatively modest.
  3. I would like to know if the authors have found an answer to the question of how the described models are applied to different breeds of dairy cattle and what are the influences on their use in different climatic zones.
  4. Is the size of farms and their economic situation respected in the mathematical models and how is it reflected in the decisive criteria and their speed of herd renewal?
  5. For many farms, the indicator of maintaining favorable conditions for dairy cows (secured animal welfare) is the number of their productive years achieved. There are significant differences in this. Was this criterion taken into account?
  6. In part 3. Results and Discussion, the authors analyze the found and processed data from the scientific literature. From a formal point of view, I recommend editing the English title of the text in Figure 4. on line 196. Some English terms in Table 3. need to be corrected, e.g. Number Publication in Paper Nuevo; Stable
  7. In the last part Conclusions, I lack a final evaluation that would indicate the benefit of applying new methods and decide which method or methods are most effective for this task.

Author Response

Dear reviewer, thank you very much for your comments. Your suggestions have provided a valuable opportunity for improvement. We proceed to answer your questions:

The authors rely on 59 scientific publications listed in the final References in the article. Given the wide range of problems that affect herd turnover, I consider this number to be relatively modest.

On lines 178-185 we answer this question.

I would like to know if the authors have found an answer to the question of how the described models are applied to different breeds of dairy cattle and what are the influences on their use in different climatic zones.

On lines 367-373 we answer this question.

Is the size of farms and their economic situation respected in the mathematical models and how is it reflected in the decisive criteria and their speed of herd renewal?

On lines 374-377 we answer this question.

For many farms, the indicator of maintaining favorable conditions for dairy cows (secured animal welfare) is the number of their productive years achieved. There are significant differences in this. Was this criterion taken into account?

On lines 378-383 we answer this question.

In part 3. Results and Discussion, the authors analyze the found and processed data from the scientific literature. From a formal point of view, I recommend editing the English title of the text in Figure 4. on line 196. Some English terms in Table 3. need to be corrected, e.g. Number Publication in Paper Nuevo; Stable

The corrections are made in lines 211 and 212 and in table 3.

In the last part Conclusions, I lack a final evaluation that would indicate the benefit of applying new methods and decide which method or methods are most effective for this task.

On lines 466-490 we answer this question.